# Land Use Multifunctions in Metropolis Fringe: Spatiotemporal Identification and Trade-Off Analysis

Linlin Wang, Qiyuan Hu, Liming Liu  and Chengcheng Yuan *

College of Land Science and Technology, China Agricultural University, Beijing 100193, China
* Correspondence: ycc@cau.edu.cn

**Abstract:** As the transition zone between urban and rural, the metropolis fringe is an area where various functions permeate and compete fiercely with each other. Understanding land use functions (LUFs) and their relationships are crucial for both urban and rural sustainable development. In this study, we established a conceptual framework of land use multifunctions in the urban fringe and proposed an improved evaluation method to quantify LUFs at the grid scale. The bivariate spatial autocorrelation method was used to explore the trade-offs among LUFs. Taking Qingpu District in Shanghai as a case study, we found that LUFs displayed pronounced spatiotemporal heterogeneity. The economic- and social-dominated functional trade-off mainly occurred in the east part of Qingpu, whereas the ecological function dominated in the west. Human preference and corresponding policies were the key factors leading to these trade-offs. Additionally, land use function zoning was proposed to resolve existing conflicts. These findings can provide scientific information for efficient land use management in the metropolis fringe.

**Keywords:** land use functions; function evaluation; trade-offs; grid scale; metropolis fringe



## 1. Introduction

China has witnessed unprecedented urbanization over the past few decades, which is characterized by rapid urban growth [1,2]. From 1978 to 2017, the ratio of the urban population rose from 17.9% to 58.5% and, consequently, the built-up areas expanded by 0.48 million hectares. As the frontier of urban expansion [3], urban fringe, especially metropolis fringe, is the most dynamic area in the course of urbanization [4]. Rapid urban growth has led to the continuous consumption of farmland and ecological land. Despite significant economic growth, excessive land use has caused a variety of environmental and social issues [5], such as water pollution [6], biodiversity loss [7], forest degradation [8], and food insecurity [9], which pose a serious threat to sustainable development [10].

Facing the deterioration of the human–land relationship, policymakers and scientists have gradually realized that economic-output-oriented land use patterns are unsustainable and have begun to change their land use and management strategies [11,12]. In 2007, the Chinese government first proposed the strategic objective of ecological civilization construction. To pursue sustainable development, the central government has repeatedly emphasized the establishment of a territorial spatial planning system since 2013. Therefore, a new policy tool known as the "three zones and three lines" was proposed to balance economic development, food security, and ecological protection and to optimize the layout of territorial space [8]. From the national to township level, in the transformation from one single economic function, land use management is required for coordinating various functions to maximize the overall benefits [13,14]. Situated in the transition zone between urban and rural areas, the urban fringe is the region where "three lines" meet [15]. Conflicts among multiple functions are most pronounced in this area. Furthermore, the promotion of sustainable land use is a tricky problem facing the metropolis fringe.

The key to creating a sustainable land use pattern is to identify and assess the major function of each area and explore the interactions among multiple functions. Multifunctionality research not only establishes a solid theoretical foundation, but also provides an important approach for this study [16–18]. The concept of multifunctionality originates from the agriculture sector, which emphasizes that agriculture provides various services besides food production [19]. In the context of sustainable development, this concept has been introduced into the field of land use and has become a new paradigm for land science research [20]. Land is the carrier upon which mankind relies for existence and provides diverse products and services to human society, collectively known as land use functions (LUFs) [21]. Under the pressure of population growth and economic development, human beings have long pursued the economic output of land use, aggravating the unbalance among multiple functions. Multifunctional land use aims to satisfy human development needs as much as possible and minimize the cost to the ecological environment. Thus, it is considered to be an important method for alleviating the existing conflicts and achieving sustainable development [17,22,23].

Recently, land use multifunctionality research at the national or regional scale has gained increasing academic attention [13,24,25], but such research in the metropolis fringe is still in its infancy. In other words, it remains unclear how the functions are distributed and affected by each other in the urban fringe. In addition, previous studies associated with the evaluation of LUFs usually took administrative cells as their basic unit [26], but they cannot meet the requirements of land use management in this relatively small study area. Therefore, it is necessary to explore the characteristics of LUFs on a more detailed scale to pointedly and scientifically guide land use practices. More critically, the impact of rapid urbanization on LUFs is an ongoing process. However, few researchers have gained insight into the dynamic assessment of LUFs and their path of evolution, and, thus, the spatial and temporal variation in LUFs and their relationships have not been revealed.

As one of the most rapidly growing cities in China, Shanghai has been experiencing dramatic urban sprawl and is confronted with various land use problems. Thus, an empirical study was conducted in Qingpu District, Shanghai, aiming at the following: (1) quantifying and spatializing the LUFs at grid scale, and revealing the spatiotemporal change characteristics of LUFs; (2) analyzing the relationships among various functions under the influence of urbanization; and (3) proposing a land use function zoning scheme, based on the above research, to provide a reference for spatial planning. As a typical representative of urban fringe, our study may be valuable as a reference for the sustainable land use of many regions with similar backgrounds.

## 2. Materials and Methods

### 2.1. Study Area

Qingpu District is located southwest of Shanghai, bordering Zhejiang Province and Jiangsu Province. It lies between 120°53′ E–121°17′ E and 30°59′ N–31°16′ N (Figure 1), covering a total area of 668.52 km². The region includes 3 subdistricts and 8 towns, with a total of 184 administrative villages. Qingpu is an important ecological conservation area, with the largest freshwater lake, Dianshan Lake of Shanghai, in the west. However, the ecological environment is experiencing a crisis due to human activities.

Located at the fringe of the international metropolitan Shanghai, Qingpu is experiencing rapid industrialization and urbanization. The gross domestic product (GDP) of Qingpu increased from RMB 12.54 billion in 2000 to RMB 100.92 billion in 2017. Additionally, the secondary sector is still the main engine of local economic growth, which has led to a range of environmental issues, including air pollution, water pollution [6], and land pollution. In 2017, the population of Qingpu was 1.21 million, a 102% increase from 2000. Urban sprawl is encroaching on the agricultural space and ecological space, resulting in a conflict between economic development, crop production, and environmental protection [27]. Determining how to coordinate different functions has become a challenge in this area, making Qingpu a typical metropolis fringe and a good location for conducting multifunctionality research.

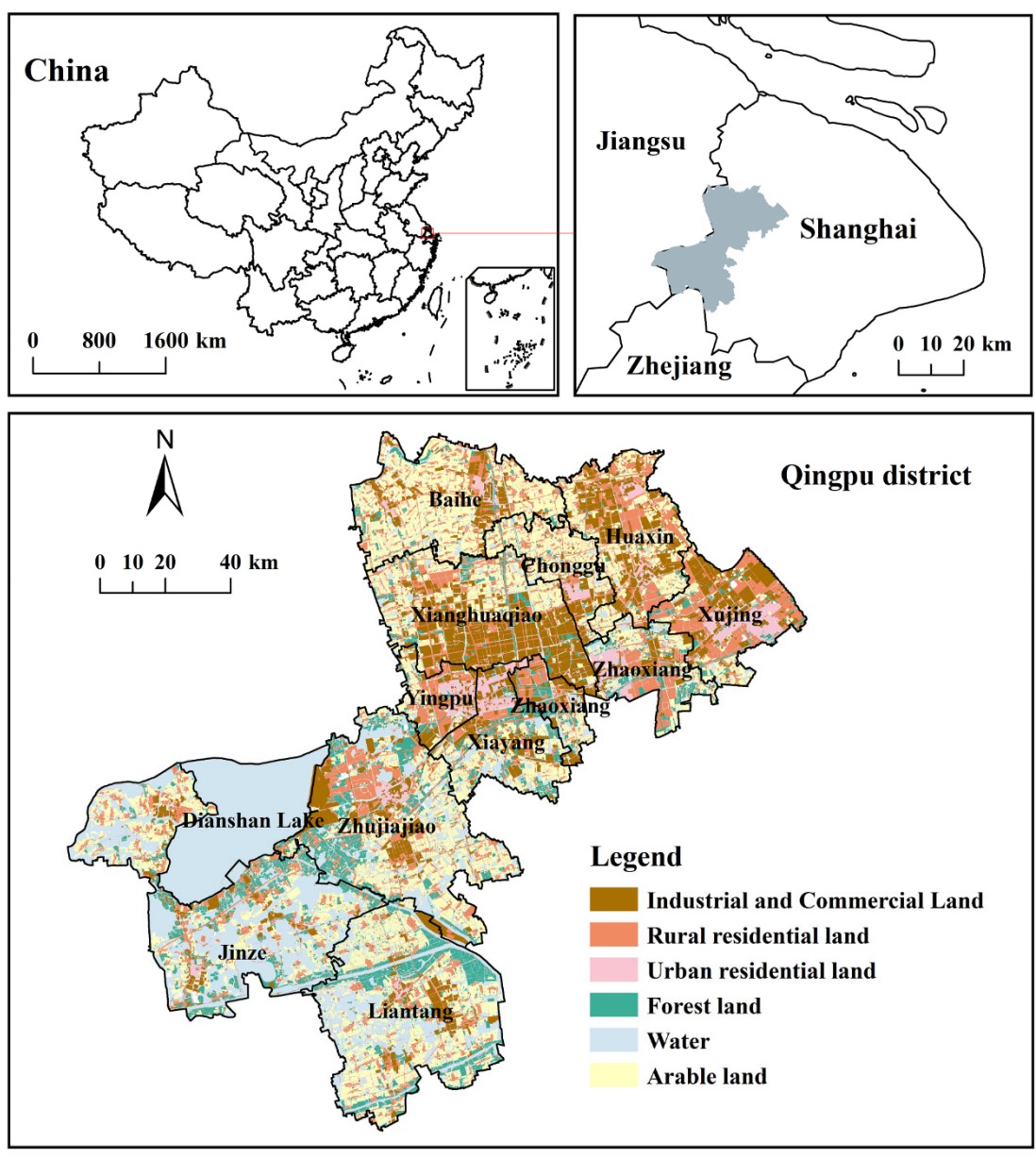

**Figure 1.** Location of Qingpu District in Shanghai.

*2.2. Data Sources and Processing*

This paper focuses on the evaluation of LUFs using spatial and statistical data. The land use/land cover data from 2000, 2009, and 2017 were obtained through the interpretation of the Landsat Enhanced Thematic Mapper (ETM) or Operational Land Imager (OLI) images at a 30 m resolution, collected from the Geospatial Data Cloud Platform (http://www.gscloud.cn/, accessed on 5 November 2022). The overall accuracy of classification was over 90%, meeting the research requirements [28]. Land use in this study was classified into six categories: forest, water bodies, arable land, urban residential land, rural residential land, and industrial and commercial land. The Normalized Difference Vegetation Index (NDVI) was the product of MOD13Q1, with a 250 m spatial resolution and 16 d time resolution, and was acquired from the Land Processes Distributed Active Archive Center (LP DAAC) (http://lpdaac.usgs.gov/main.asp, accessed on 5 November 2022). We selected 10 images from the critical growing period (from May to September) of each year, and the NDVI data of 2000, 2009, and 2017 were obtained by using the maximum synthesis method [29].

Socioeconomic data such as crop yield, GDP, and population on a village scale were obtained from the Qingpu Statistical Yearbook. All spatial data were ultimately rescaled to a 250 m × 250 m grid as the basic evaluation unit using a Gauss–Kruger projection and the Xi'an 80 geographical coordinate system.

*2.3. Methods*

2.3.1. Conceptual Framework of Land Use Multifunctions (LUFs) in the Urban Fringe

A land use system is a typical, complex human–Earth system, and LUFs are the products of human–land interactions (Figure 2). In other words, human beings transform and utilize land resources according to their survival and development needs, which in turn produces diversified products and services for humans. In most developing countries, metropolitan fringe areas play an important role in national economic growth. Thus, both central and local governments tend to expand metropolitan fringe areas. As located in the transition zone between urban and rural areas, the socioeconomic structure of the urban fringe is obviously dualistic. Thus, in the conceptual framework (Figure 2), the land use structure is been highlighted to be an internal factor of multifunctional land use in the urban fringe.

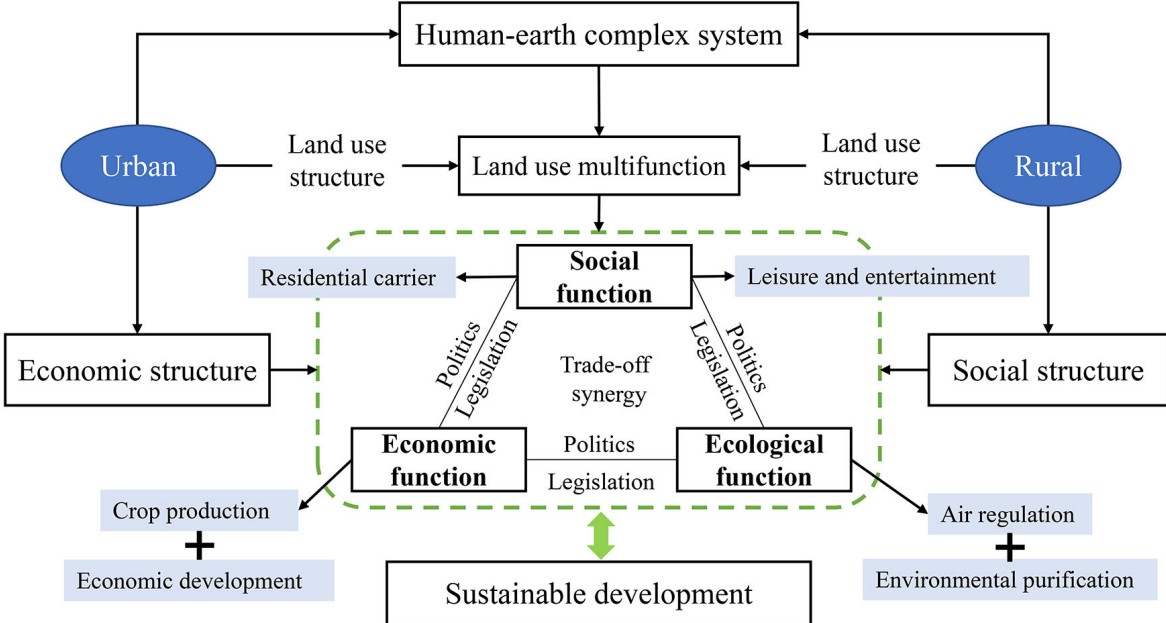

**Figure 2.** The conceptual framework of LUFs in the urban fringe.

For the foundation and conceptual framework of LUFs in the urban fringe, it is appropriate that land use should not only meet the needs of industrial and commercial development, but also provide living and leisure space for urban and rural residents. The LUFs are specified as the social function, economic function, and ecological function. Ecological functions refer to the ability to provide nonliving resources, maintain biodiversity, and regulate the ecological environment [17]. In the conceptual framework, ecological functions mainly include the air regulation function and environmental purification function. Economic functions are the basis for maintaining human survival through material production and economic development, including the crop production function and economic development function. Social functions are the ultimate goal of land use, as they satisfy the physical and mental requirements of human beings [30–32]. The social functions mainly consist of the residential carrier function and leisure and entertainment function. The three land use functions are traded off and synergized through local politics and legislation. This conceptual framework considers the adequate use of land resources, which is essential for the sustainable development of urban and rural areas.

In trying to satisfy the increasing and sometimes contradictory needs of both urban and rural residents, land use tends to be complicated. To be more specific, cultivated land is still the main land use type in this region, providing subsidiary agricultural products. Additionally, a large amount of natural space makes this area an important ecological shelter for the central city. Meanwhile, with the acceleration of urbanization, land use should not only meet the needs of industrial and commercial development, but also provide living and leisure space for urban and rural residents. Consequently, the urban fringe is the region where various functions coexist and compete fiercely with each other. To coordinate various functions and actualize sustainable land use, this paper adopted the "economy–society–ecology" classification framework [12,20], and established an evaluation index system according to the actual situation of the study area. There are complex relationships among the ecological, economic, and social functions. The goal of land use is to achieve the coordinated development of multiple functions, ensuring that land resources can continuously provide products and services for humans.

2.3.2. Quantification of LUF Index

To obtain LUF information for making more accurate and effective decisions about urban fringe land use, relevant spatial models were used to quantify and spatialize the indicators from 2000 to 2017 at a 250 m grid scale in the ArcGIS 10.2 software.

The crop production function is the ability to provide agricultural products for the region, which can be measured by crop yield per unit area. Previous studies have shown that there is a significant linear relationship between crop yield and the NDVI [30]; thus, we spatially allocated crop production statistics reported on the village scale according to the maximum NDVI between May and September. The economic development function is the region's output capacity of nonagricultural economic activities, and the economic output per unit area was selected to reflect the economic development level of the region. GDP distribution is closely related to land use type and industrial development level [33]. By establishing the correlation between industrial output value and land use data, a spatial model of the economic development function was constructed, as shown in Table 1.

The residential carrier function refers to the region's ability to accommodate the population and provide living space for residents, which is represented by the population density index. The leisure and entertainment function refer to the ability to provide human beings with places for viewing, tourism, and leisure, thus enabling them to obtain psychological satisfaction and spiritual enjoyment. It can be characterized by the tourist attraction accessibility index, that is, the time and distance from any grid to the nearest tourist attraction [34]. The longer the time needed, the worse the accessibility and the lower the leisure and entertainment function value.

Considering data availability and operability, this paper adopted the adjusted ecosystem service value equivalent factor per unit area and ecosystem service value coefficient [35] to spatially simulate the air regulation function and environment purification function of different types of ecosystems; the calculation methods are shown in Table 1.

After evaluating the LUFs, the indicators were standardized using the range standardization method (except for the indicator of tourist attraction accessibility, the other indicators were all positive) so that the values of all indicators ranged from 0 to 1. Moreover, all subfunctions were assigned the same weight since they are equally important for regional development. Three primary function value indices were calculated with the weighted sum method.

**Table 1.** Land use functions, indicators, and quantification methods.

| Primary Functions | Subfunctions | Indicators | Formula | Formula Description |
|---|---|---|---|---|
| Economic function | Crop production | Per unit area crop yield | $Grain_i = \frac{NDVI_i}{NDVI_j} \times Grain_j$ | $Grain_i$ and $Grain_j$ are the crop yields for grid $i$ and village $j$, and $NDVI_i$ and $NDVI_j$ are the NDVIs of grid $i$ and village j, respectively. |
| | Economic development | Per unit area output value | $Econ_i = \sum a_i x_i + \sum b_i y_i$ | $Econ_i$ is the economic output of grid i. $a_i$ is the GDP distribution coefficient of arable land, forest land, water area, and rural residential land in grid $i$, which can be obtained by dividing the added value of agriculture, forestry, fishing, and sideline activities of each village by the area of arable land, forest land, water area, and rural residential land, respectively. $x_i$ is the corresponding area percentage of each land use type in grid $i$. $b_i$ is the GDP distribution coefficient of industrial and commercial land in grid $i$, which can be obtained by dividing the added value of the second and third industries by the area of industrial and commercial land. $y_i$ is the proportion of industrial and commercial land in grid i. |
| Social function | Residential carrier | Population density | $Pop_i = \frac{Pop_j}{R_j} \times R_i$ | $Pop_i$ and $Pop_j$ are the resident populations for grid $i$ and village $j$, and $R_i$ and $R_j$ are the residential land areas of grid $i$ and village $j$, respectively. |
| | Leisure and entertainment | Accessibility of tourist attractions | $K_i = \begin{cases} \frac{1}{2} \sum\limits_{i=1}^{n} (C_i + C_{i+1}) \\ \frac{\sqrt{2}}{2} \sum\limits_{i=1}^{n} (C_i + C_{i+1}) \end{cases}$ | $K_i$ is the time to arrive in the nearest tourist attraction from grid $i$; $C_i$ is the cost value of grid $i$; $C_{i+1}$ is the cost value of grid $i + 1$ along the direction of motion; and $n$ is the total number of grids. The upper fraction represents the time cost of the grid along the vertical or parallel direction, and the lower fraction represents the time cost along the diagonal direction. The time cost of an expressway, national road, provincial road, county road, other road, and land surface is 0.125, 0.1875, 0.25, 0.375, 0.5 and 3 min, respectively. |
| Ecological function | Air regulation | – | $AR_i = \sum A_i VC_i$ | $AR_i$ is the air regulation function value of grid $i$, $A_i$ is the area of each type of ecosystem in grid $i$, and $VC_i$ is the corresponding ecosystem service value coefficient in grid i. |
| | Environmental purification | – | $EP_i = \sum A_i VC_i$ | $EP_i$ is the environmental purification function value of grid $i$, $A_i$ is the area of each type of ecosystem in grid $i$, and $VC_i$ is the corresponding ecosystem service value coefficient in grid i. |

2.3.3. Trade-Off and Synergy Analysis

There are complex trade-offs and synergies between LUFs [36]. Trade-off is the increasing supply of one function at the expense of another. Synergy means that the increase or decrease in one function will lead to the increase or decrease in another function; in other words, the two functions have the same changing tendency [37,38]. To explore the spatiotemporal patterns of the trade-off and synergy of LUFs, a bivariate spatial autocorrelation analysis [39], which is widely used in ecosystem services [40,41], was carried out.

Bivariate global spatial autocorrelation is used to test the spatial correlation degree of two attributes of the spatial unit, which can be measured by the global Moran's I tool using the following formula.

$$I_{UL} = \frac{n \sum_{i=1}^{n} \sum_{j=1}^{n} W_{ij} \left( \frac{X_i^U - \overline{X_U}}{\sigma_U} \right) \left( \frac{X_j^L - \overline{X_L}}{\sigma_L} \right)}{(n-1) \sum_{i=1}^{n} \sum_{j=1}^{n} W_{ij}} \tag{1}$$

$$Z(I) = \frac{1 - E(I)}{\sqrt{Var(I)}} \tag{2}$$

where $I_{UL}$ represents the bivariate global spatial autocorrelation coefficient. The range of Moran's $I$ is $[-1, 1]$. $X_i^U$ is the value of the Uth land use function of grid $I$ and $\overline{X_U}$ is the average value of the Uth land use function. n is the total number of grids and $\sigma$ notes variance. $W_{ij}$ is the spatial weight matrix calculated by queen contiguity. $E(I)$ is the mathematical expectation. $Z(I)$ is a test value, whereby there is an extremely significant spatial correlation at the 5% level of probability when $|Z(I)| \geq 1.96$ [40]. At the given significance level, when Moran's $I > 0$, there is a positive spatial autocorrelation, indicating that the LUFs had a spatially significant synergy relationship. When Moran's $I < 0$, there is a negative spatial autocorrelation, indicating that the LUFs had a spatially significant trade-off relationship. When Moran's $I = 0$, there is no spatial autocorrelation.

The bivariate local spatial autocorrelation index can reveal the degree of correlation between one attribute value of each spatial unit and another attribute value of the adjacent spatial unit. Local Moran's I statistics are used for measurement, and the calculation formula is as follows:

$$I_i^{UL} = \frac{X_i^U - \overline{X_U}}{\sigma^U} \sum_{j=1}^{n} \left( W_{ij} \frac{X_j^L - \overline{X_L}}{\sigma^L} \right) \tag{3}$$

where $I_i^{UL}$ represents the bivariate local spatial autocorrelation coefficient of grid *i*. Based on the local Moran statistics, a local analysis is visualized in the form of cluster maps. The relationship between functions can be divided into an HH (high–high) synergy region, LL (low–low) synergy region, HL trade-off region (high–low), LH trade-off region (low–high), and functional compatibility relationship region (NS).

## 3. Results

### 3.1. Evolution Trends and Spatial Distribution of Land Use Subfunctions

Spatial patterns of selected LUFs exhibited pronounced heterogeneity and greatly changed over time (Figure 3). The results (Figure 3(c1–c3)) show that the center of the crop production function gradually shifted from the northern and eastern region of Qingpu to the western region. At the town level, all other subdistricts and towns except for Liantang were in a degraded state, which was especially true in economically developed areas such as Xujing, Huaxin, and Zhaoxiang, where this function reduced by 80.9%, 80.5%, and 67.5%, respectively, as compared with data from 2000. On the contrary, the economic development function was greatly improved, gradually changing from a scattered to spatial concentration (Figure 3(b1–b3)) during this period. Functional growth areas mainly concentrated in Xianghuaqiao and Huaxin contributed 99.5% of total growth. Jinze, Liantang, and Zhujiajiao in the west experienced declines in economic development function due to their strict environmental protection policies.

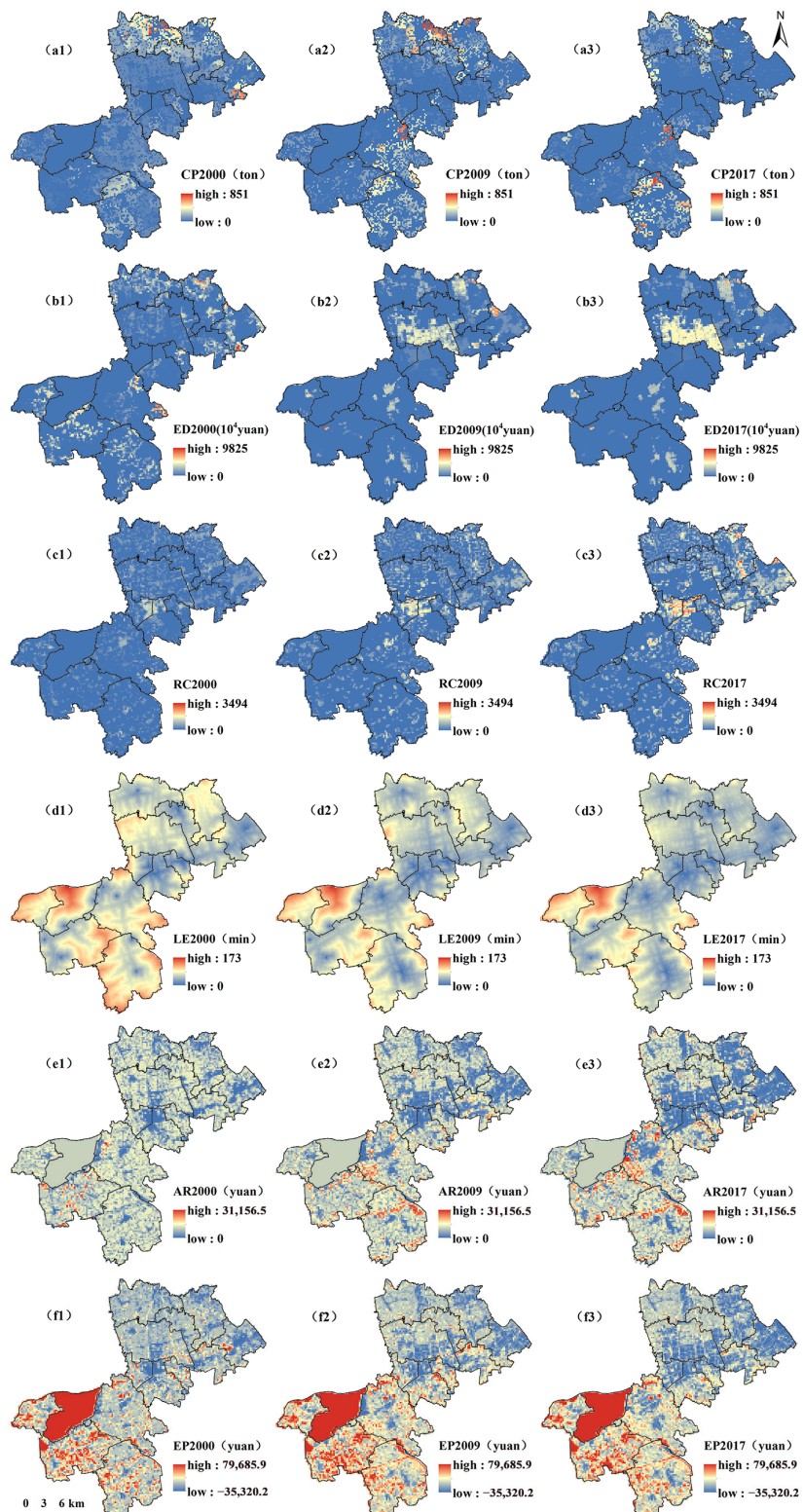

**Figure 3.** Spatial distribution of land use subfunctions in Qingpu District from 2000 to 2017. The land use subfunctions include crop production (CP) in 2000 (**a1**), 2009 (**a2**) and 2017 (**a3**); economic development (ED) in 2000 (**b1**), 2009 (**b2**) and 2017 (**b3**); residential carrier (RC) in 2000 (**c1**), 2009 (**c2**) and 2017 (**c3**); leisure and entertainment (LE) in 2000 (**d1**), 2009 (**d2**) and 2017 (**d3**); air regulation (AR) in 2000 (**e1**), 2009 (**e2**) and 2017 (**e3**); environmental purification(EP) in 2000 (**f1**), 2009 (**f2**) and 2017 (**f3**).

The residential carrier function (Figure 3(c1–c3)) increased significantly and showed similar spatial distribution patterns when compared to the economic development function. Higher values were observed in the center of Qingpu and for every town, and gradually reduced from the center to outer areas. Meanwhile, Huaxin, Xujing, Xiayang, and Zhaoxiang in eastern Qingpu had the largest proportion (63.0%) of the total growth in the residential carrier function. In terms of the leisure and entertainment function (Figure 3(d1–d3)), the majority of towns had an increase over the 17 years. This was especially seen in Liantang, Huaxin, and Zhujiajiao, where this function increased by 46.3%, 39.3%, and 29.1%, respectively, since 2000, accounting for 57.9% of the total growth.

The air regulation function was generally weakened, but the spatial distribution was quite varied among regions (Figure 3(e1–e3)). The air regulation function in the western region presented an upward trend, which was significantly related to the increased vegetation coverage. Continuous decreases were observed in the central and eastern regions of Qingpu, especially in Xujing, Xianghuaqiao, and Huaxin, accounting for 84.1% of the total reduction. The value of the environmental purification function climbed from 2000 to 2009, but dropped from 2009 to 2017 (Figure 3(f1–f3)). The environmental purification function followed a similar spatial pattern as the air regulation function, whereby higher values existed in the west, and lower values in the center and east.

### 3.2. Spatiotemporal Variation of Primary Functions

The spatial distribution characteristics of economic function (Figure 4(a1–a3)) in Qingpu varied widely. Functional growth areas were mainly located in Xianghuaqiao, Liantang, and the east part of Zhujiajiao. The former was primarily due to the economic growth stimulated by the development of secondary and tertiary industries, whereas the latter two were attributed to the improvement in the crop production function. The degradation areas were concentrated in Baihe, Huaxin, and Xujing, which was mainly due to the loss of farmland to sprawling cities and the decline in agricultural productivity.

The social function (Figure 4(b1–b3)) improved steadily during the study period, following the spatial pattern of a gradual reduction from the town center or main traffic arteries to outer areas. Additionally, the values were generally higher in the east and lower in the west due to the different levels of economic development and infrastructure conditions.

As for the ecological function (Figure 4(c1–c3)), the high-value areas were primarily distributed in the west, especially in Dianshan Lake and its surrounding forest. Additionally, lower values existed in the center of Qingpu and the regions close to the central city of Shanghai, which declined significantly from 2000 to 2017.

### 3.3. Spatial Pattern and Variation of Trade-Off and Synergy of LUFs

Table 2 shows that the global Moran's I meets the test standards at the 5% level of probability, indicating that the LUFs had a significant spatial correlation. Additionally, most of the functions had a spatially significant trade-off relationship since the data were negative. On this basis, a bivariate local autocorrelation analysis was conducted to further explore the spatiotemporal pattern of three pairs of primary functions; the results are presented at the grid level in Figures 5 and 6.

**Table 2.** Bivariate global Moran's I indices ($p < 0.05$) and the Z-scores of land use primary functions in Qingpu.

| Function Type | 2000 | | 2009 | | 2017 | |
|---|---|---|---|---|---|---|
| | **Moran's I** | **Z-Score** | **Moran's I** | **Z-Score** | **Moran's I** | **Z-Score** |
| EF–SF | 0.035 | 7.545 | −0.028 | −8.566 | −0.050 | −19.908 |
| EF–ECF | −0.100 | −28.179 | −0.086 | −22.620 | 0.006 | 2.032 |
| SF–ECF | −0.418 | −112.669 | −0.400 | −90.678 | −0.422 | −79.867 |

Note: EF—economic function; SF—social function; ECF—ecological function.

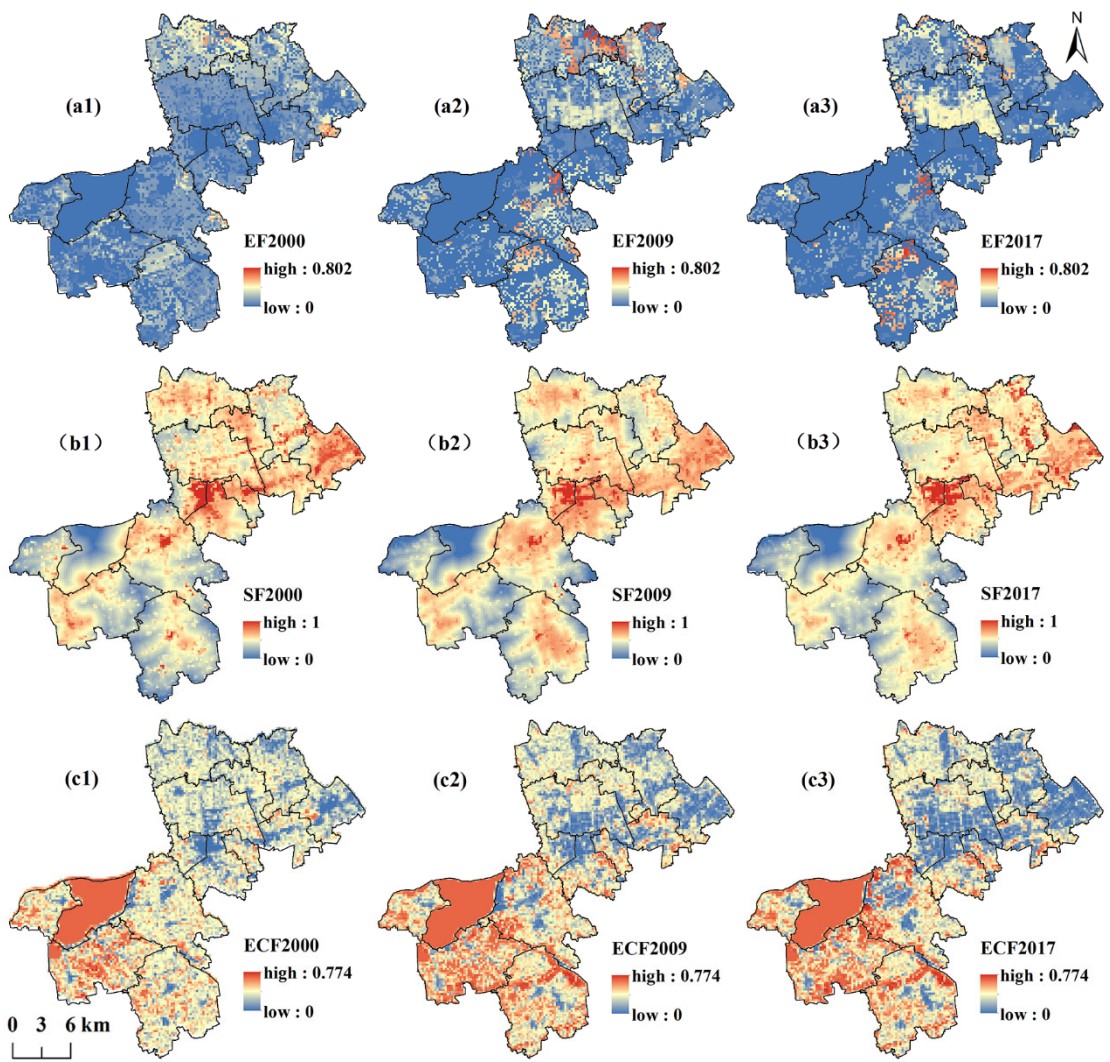

**Figure 4.** Spatial distribution of land use primary functions in Qingpu District from 2000 to 2017. The land use primary functions include three functions: economic function (EF) in 2000 (**a1**), 2009 (**a2**) and 2017 (**a3**); social function (SF) in 2000 (**b1**), 2009 (**b2**) and 2017 (**b3**); ecological function (ECF) in 2000 (**c1**), 2009 (**c2**) and 2017 (**c3**).

There were obvious differences in the trade-off and synergy spatial pattern (Figure 5) and quantity (Figure 6) among functions. The global Moran's I of EF–SF changed from 0.035 in 2000 and $-0.028$ in 2009 to $-0.050$ in 2017, indicating that the synergy relationship between functions turned into a trade-off relationship. Thus, areas of synergy reduced from 24.2% to 19.3% (Figure 6). Additionally, trade-off areas showed an increasing and then declining trend, accounting for 21.2%, 26.3%, and 20.8% of the total area in 2000, 2009, and 2017, respectively. Specifically, LH trade-off areas were concentrated in the central and eastern regions, whereas the HL trade-off areas were scattered across the western regions.

The global Moran's I of EF–ECF ranged from $-0.100$ to 0.006, indicating that the trade-off relationship between functions transformed into a weak synergy relationship. However, the spatial trade-off between EF–ECF was more evident, and the areas of trade-off increased from 19.9% to 27.9%. HL trade-off areas were observed in the central and eastern regions, gradually changing from scattered to concentration. LH trade-off areas were mainly distributed in the western region.

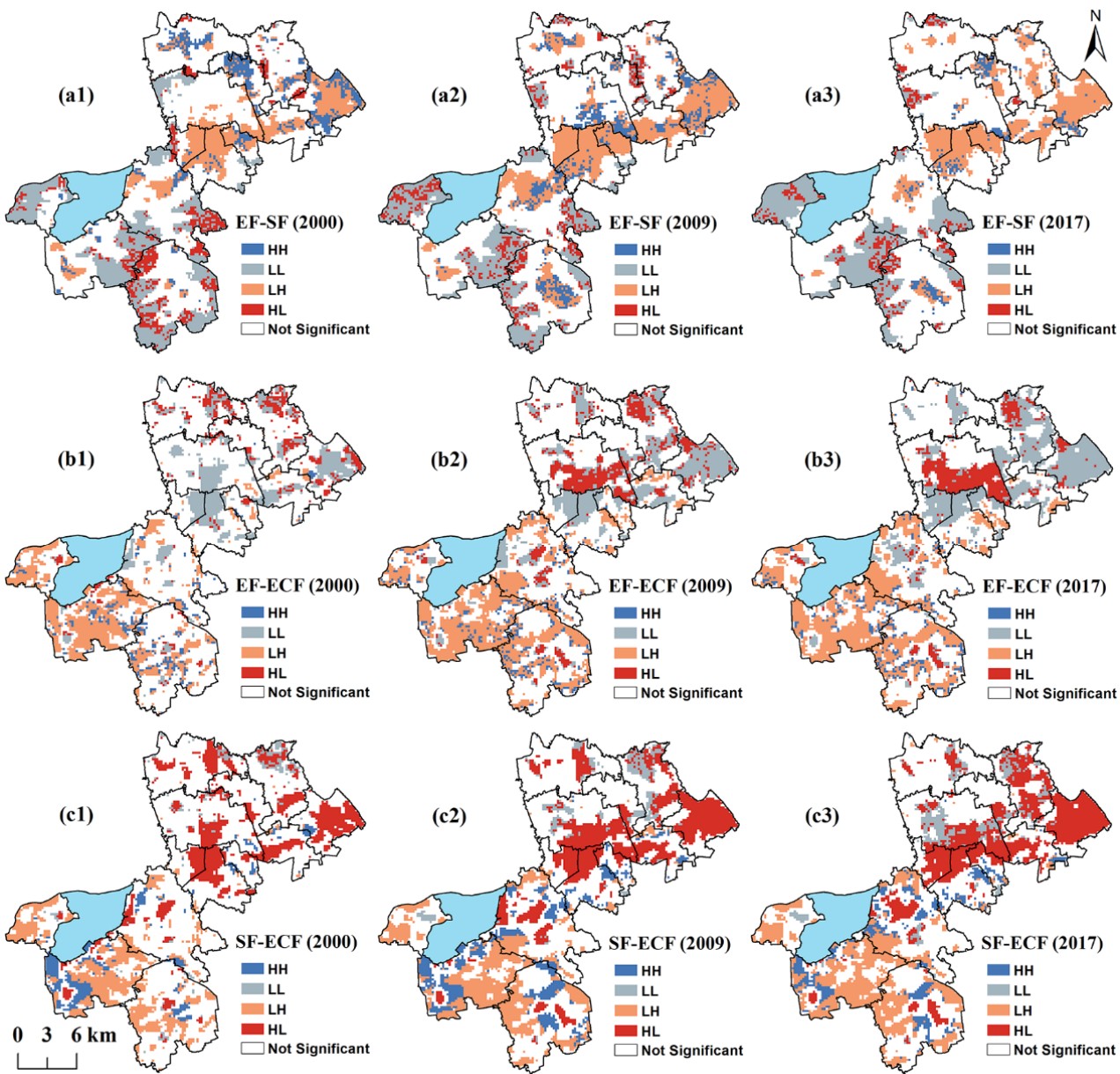

**Figure 5.** Spatial synergy and trade-off of land use primary functions in Qingpu District during 2000–2017. The results of spatial synergy and trade-off include economic function (EF) and social function (SF) in 2000 (**a1**), 2009 (**a2**) and 2017 (**a3**); economic function (EF) and ecological function (ECF) in 2000 (**b1**), 2009 (**b2**) and 2017 (**b3**); social function (SF) and ecological function (ECF) in 2000 (**c1**), 2009 (**c2**) and 2017 (**c3**).

The global Moran's I of SF–ECF was −0.418, −0.400, and −0.422, showing a strong trade-off relationship and trend of decreasing first and then increasing. The trade-off areas expanded continuously and increased from 28.7% in 2000 to 38.5% in 2017. Specifically, HL trade-off areas were concentrated in the central and eastern regions, whereas the LH trade-off areas were mainly distributed in the western regions.

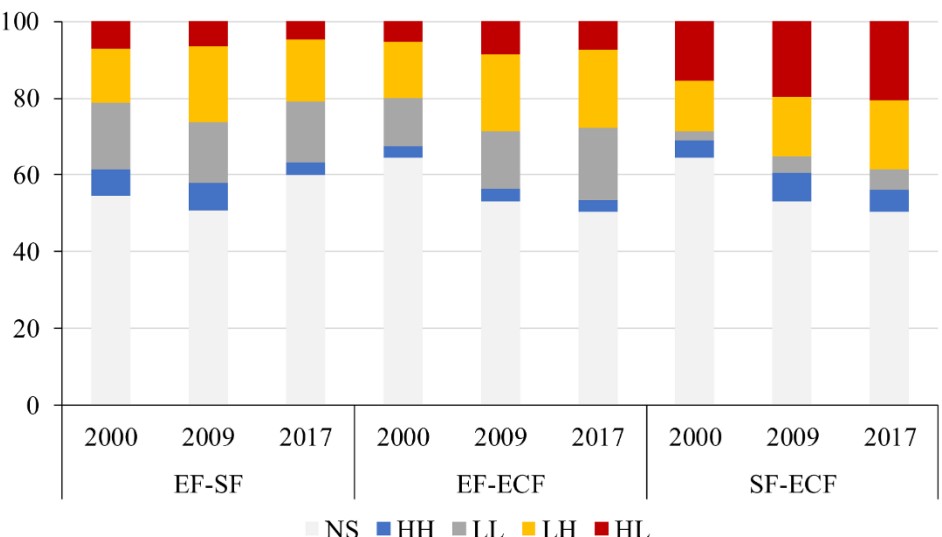

**Figure 6.** The proportion of LUF trade-off and synergy in Qingpu District during 2000–2017 (EF—economic function; SF—social function; ECF—ecological function).

## 4. Discussion

### 4.1. Mechanism Analysis of Functional Conflicts

To achieve the coordinated development of multiple functions, we first need to understand the mechanisms behind LUF trade-offs or conflicts [42]. Land use is essentially a process in which human beings utilize and transform land resources according to their needs [32]. Human activities always involve land use change, which in turn affects the land's ability to provide products and services for humans [43]. The arising LUF conflicts come from the human preference for different functions [13]. Specifically, when the consumption of land for a specific function is maximized, the supply of land for other functions will be intentionally or unintentionally weakened. For example, in order to increase yields, a huge amount of chemical fertilizer and pesticide was used, causing soil degradation and water pollution. This preference is considered when formulating new policies, which further intensifies the conflicts between LUFs [44].

In terms of time scale, human needs vary at different socioeconomic development periods, according to Maslow's hierarchy of needs [45]. Thus, LUFs and their relationship change over time. Qingpu was experiencing rapid industrialization and urbanization from 2000 to 2017 (Figure 7), and economic growth was the primary target of regional development. Namely, the economic function was given priority over the ecological function and social function. Therefore, the economic function and ecological function were always in conflict during the whole study period.

Specifically, the research period can be divided into two stages according to per capita GDP and industrial structure. The middle stage of industrialization was from 2000–2007, and the late stage of industrialization was from 2008–2017. In the first period, economic growth was the main goal of regional development, and less attention was given to the ecological environment. By the late industrialization stage, residents' demands began to shift to ecological goods as their life quality had improved dramatically and material needs had been met to a certain extent. In order to maintain and restore the ecological function, Shanghai and Qingpu have issued a series of policies and gradually increased their investment in environmental protection. For instance, the Qingpu District Government published the Notice on the Implementation Opinions of Public Welfare Forests Construction in 2004, focusing on the construction of water conservation forests around Dianshan Lake and the upper Huangpu River. Consequently, the forest land greatly increased to over ten thousand hectares at the end of 2017, and the forest coverage increased from 2.6% in 2000 up to 13.8% in 2017. Meanwhile, the Shanghai Municipal

Government promulgated the Ecological Compensation Transfer Payment Measures in 2009, which established an ecological compensation system for water source areas, basic farmland, and public welfare forests, and provided financial support for Qingpu District's environmental protection work. Furthermore, Shanghai introduced a construction land reduction policy in 2013 to dismantle low-efficiency industrial land and to improve the local environmental quality by transforming it into agricultural or ecological land and reducing pollution sources [13]. According to the Qingpu Master Plan and Land Use Master Plan (2017–2035), making land ecological and livable has been the principal strategy for regional development. The ecological function of Qingpu improved as the series of policies and measures were adopted. Therefore, the intensity of conflict between the economic development function and the ecological function climbed from 2000 to 2009, but dropped from 2009 to 2017.

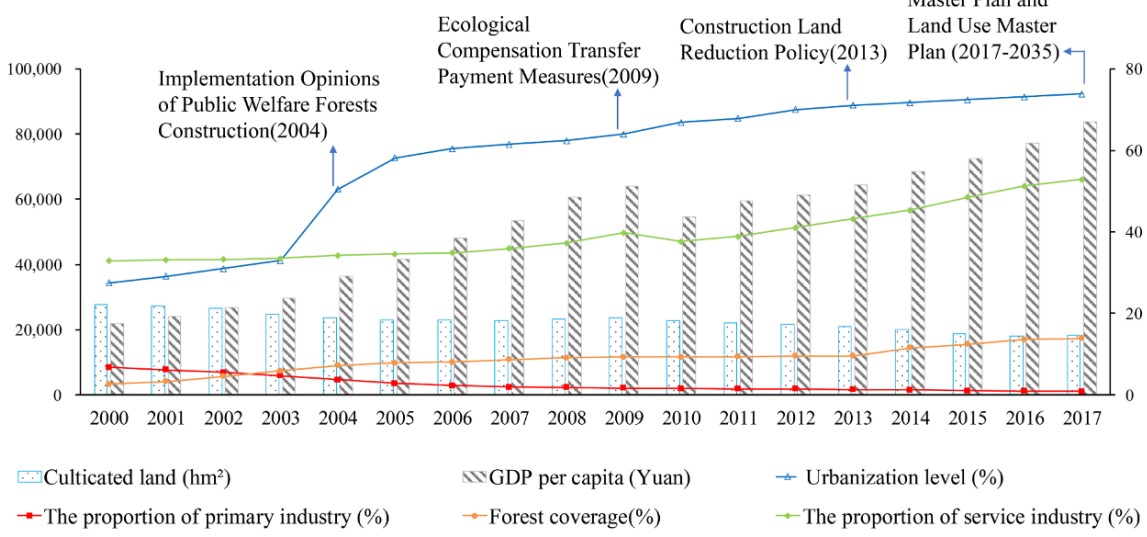

**Figure 7.** Summary of land policies and changing socioeconomic indicators in Qingpu District from 2000 to 2017.

In terms of spatial scale, there were significant differences in the LUFs and functional conflicts among regions due to the varied resource endowments and the background context. The economic development function of eastern Qingpu, which is adjacent to central Shanghai, is higher than that of the western part. At the same time, the residential carrier function has greatly improved, as eastern Qingpu has become the main inflow area for people coming to Shanghai due to its employment opportunities and relatively low cost of living. Therefore, the economic- and social-dominated functional conflicts were mainly located in the eastern part of Qingpu. Western Qingpu is located in the upstream water source protection area of the Huangpu River. Special environmental protection policies and strict land use policies have been implemented over the long term to restrict the development of secondary industries around the Dianshan Lake while controlling fish farming in natural water. The ecological environment has been extremely improved through industrial restructuring, public welfare forest construction, and pollution control. Under the constraint of ecological protection, the economic growth of the three towns in western Qingpu has been slow, whereas the ecological function has steadily improved. Accordingly, the western region showed a functional conflict dominated by the ecological function.

### 4.2. Land Use Function Zoning and Policy Implications

Land use function zoning was conducted via cluster analysis based on the trade-off analysis, as it can not only identify the dominant function, but also the major conflict types of the region, thus providing an orientation for land use management. Firstly, the proportion of each kind of trade-off and synergy in the 184 administrative villages was calculated.

Then, the study area was divided into three zones using K-means clustering modules in GeoDa software, including an ecological conservation zone, agricultural production zone, and urban development zone (Figure 8).

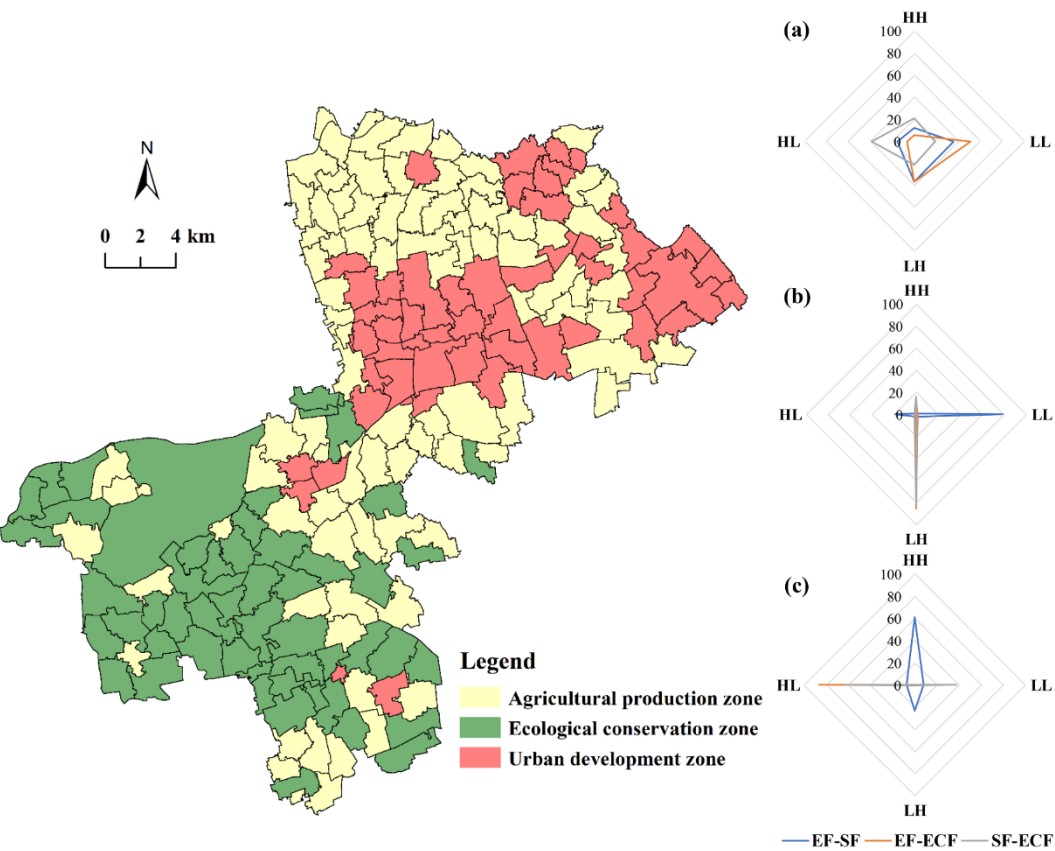

**Figure 8.** Land use function zoning in Qingpu District. Panels (**a**–**c**) show the proportion of each kind of trade-off and synergy of agricultural production zone, ecological conservation zone, and urban development zone, respectively (EF—economic function; SF—social function; ECF—ecological function).

The agricultural production zone is widely distributed in the western and northern part of Qingpu, which is an important grain and vegetable supply area of Shanghai. The relationships among multiple functions are intricate in this region (Figure 8a). Given the special location of the water conservation area, western Qingpu should actively promote environmentally friendly agricultural production, such as ecological agriculture and leisure agriculture, to ensure the gradual increase in commodity output of cultivated land, whereas the noncommodity output should remain stable or even increase [11]. At the same time, farmers' land use behavior can be guided to achieve a win–win situation for agricultural production and the ecological function through the gradual decoupling of agricultural subsidies from production and instead linking their behavior with environmental protections.

The ecological conservation zone is mainly located in the western region of Qingpu, in which the ecological function is the major function and has a trade-off relationship with the other two functions (Figure 8b). In order to alleviate the conflict between environmental protection and economic development, an ecological compensation mechanism should be further promoted to ease the financial pressure of local governments and involve more residents in protecting the environment [12]. Developing the tourist industry to rely on the specific natural and human landscape is another feasible approach. Furthermore, this region should continue to reduce the land used for inefficient construction and set strict limits on land use behavior. Additionally, it is necessary to improve the living conditions of rural residential areas by strengthening infrastructure and public service facilities without damaging the environment.

The urban development zone is mainly situated in the eastern region near central Shanghai and the center of Qingpu, which focuses on economic development and social security (Figure 8c). Given that economic growth was at the expense of the ecological environment in the long term, special attention should be given to protecting the regional ecological environment through increasing urban green space, which is also the most important and direct way to improve the urban habitat environment. At the same time, Qingpu ought to accelerate its industrial transformation and upgrading by taking advantage of the neighboring central city of Shanghai, and continuously optimize its urban functions. More importantly, compact urban patterns with intensive land use should be constructed to control the disorderly sprawl of built-up land.

*4.3. Advantages and Limitations*

Given the special location of the urban fringe, properly using land resources is crucial for the sustainable development of both urban and rural areas. In this paper, we improved the accuracy by downscaling the evaluation unit to a 250 m grid level, and applied it to our multifunctional research in the metropolis fringe. The results indicate that the assessment results based on grid cells can identify the problematic areas more precisely and assist the policymaker in executing proper land use zoning so that land use conflicts can be controlled. It can also be used as a feasible method to timely and accurately monitor LUFs and the hotspots of conflicts during the process of land use [46].

Moreover, the relationship among multiple functions at different spatial scales currently attracts much more academic attention. Our study explored the interrelationships between various functions in the grid cells at the county level, finding empirical evidence for the horizontal comparison between different studies [41]. It was found that the relationship between certain LUFs may be consistent at multiple spatial scales. According to [12], there was a trade-off relationship between the economic development function and ecological function at the national scale. Fan et al.'s [24] research showed that the crop production function and ecological function have a synergetic relationship at the provincial scale, and similar results were obtained in our study. However, some results are inconsistent with those of other scholars [27]. Therefore, further research needs to focus on the change in LUFs and their relationships under different scales, and on evaluation accuracy [47].

Although we made certain contributions, there are still some limitations and uncertainties in this study. The selection of indicators is crucially important for LUF assessment, as atypical indicators may affect the results. The results of our study are basically consistent with the reality, but whether the selected index system is optimal is still worthy of further discussion. Additionally, as we were limited by the availability of data and the spatialization of indicators, many other functions were not included in the research, such as employment functions and cultural functions. Therefore, the results of our study have certain limitations.

## 5. Conclusions

Taking Qingpu District in Shanghai as a case study area, this study employed a set of spatialization models and the bivariate spatial autocorrelation method to assess the LUFs and analyze their relationships from 2000 to 2017. The results revealed the dynamic process of LUFs and their relationships in the metropolis fringe under rapid urbanization.

At the temporal scale, the evolution trends of LUFs were quite different. The economic development function, residential carrier function, and leisure and entertainment functions exhibited an upward trend in 2000–2017, whereas the crop production function continued to marginally degrade. The air regulation function and environmental purification function showed a fluctuating trend of first rising and then falling. Additionally, the economic development function gradually replaced the crop production function as the dominant function of the area. Furthermore, the direction and strength of the relationships between LUFs have greatly changed over time. At the spatial scale, there were significant differences in

the dominant function and main conflict types among regions. The economic development function and social function were much higher in the eastern region of Qingpu, whereas the ecological function was mainly situated in the western region.

Based on evaluation results, land use function zoning was proposed as an effective way to achieve a win–win development of the economic, social, and ecological functions, which can serve as a foundation for the local government to carry out planning work [48]. Additionally, we believe that land use function zoning that includes supporting policies can effectively alleviate land use conflicts. In terms of methodology, we explored a method of identifying LUFs in a 250 m grid cell, which improved the evaluation accuracy and eliminated uncertainty in the function zoning scheme. At the same time, it was also a tool for the dynamic monitoring of regional LUFs and functional conflicts, which is of great significance in space control and sustainable land use. More critically, this study can be implemented in other cities to identify conflict areas and for balanced regional development.

**Author Contributions:** Conceptualization, C.Y.; data curation, L.W.; formal analysis, C.Y.; funding acquisition, C.Y. and L.L.; methodology, L.W.; software, L.W.; supervision, C.Y.; validation, C.Y. and L.W.; writing—original draft, L.W.; writing—review and editing, L.W. and Q.H. All authors have read and agreed to the published version of the manuscript.

**Funding:** This research was funded by the National Natural Science Foundation of China, grant number No. 42001224 and No. 41471455.

**Data Availability Statement:** Data are available from the corresponding authors upon request.

**Acknowledgments:** The authors would like to thank two anonymous reviewers for their invaluable comments that led to a much-improved manuscript. The authors also appreciate editor Cara Zhao, three Academic Editor Hongsheng Chen, Yang Xiao and Mengqiu Cao and English editor Lauren Florea for their valuable time and contribution.

**Conflicts of Interest:** The authors declare no conflict of interest.

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
