# Peer review of "Land Use Multifunctions in Metropolis Fringe: Spatiotemporal Identification and Trade-Off Analysis"

_land, doi:10.3390/land12010087_

Round 1

Reviewer 1 Report

Choice of indicators and their relative significance is hugely important here. Perhaps the authors want to discuss and review their indicators a little more.

Author Response

Response:

Thanks for your valuable comments. We have added more content on indicator selection in section 2.3.2. The detailed revisions are:

“The crop production function is the ability to provide agricultural products for the region, which can be measured by crop yield per unit area.” (line 169-170)

“The economic development function is the region’s output capacity of nonagricultural economic activities, and the economic output per unit area was selected to reflect the economic development level of the region.” (line 173-175)

“The residential carrier function refers to the region’s ability to accommodate the population and provide living space for residents, which is represented by the population density index. The leisure and entertainment function refers to the ability to pro-vide human beings with places for viewing, tourism and leisure, thus enabling them to obtain psychological satisfaction and spiritual enjoyment.” (line 179-183)

“Moreover, all subfunctions were assigned the same weight since they are equally important for regional development.” (line 193-195)

Reviewer 2 Report

The topic addressed in this article proposed for publication in Land journal is a case study of the periphery of Shanghai, specifically the Qingpu district, in order to extrapolate the results of the analysis of the urban-rural metropolitan fringe. The aim is contributing to resolve the main conflicts of existing uses in peri-urban and intrametropolitan areas.

The starting point of the proposed hypothesis is considered appropriate: to understand the "fierce competition" between land-use functions (LUFs) and their relationships that are considered crucial for sustainable urban and rural development. The authors place special emphasis on the conditioning factors: China's accelerated hyperurbanization in recent decades and agrarian multifunctionality.

It is appropriate the foundation and conceptual framework of multifunctional land use in the urban periphery understanding that land use should not only meet the needs of industrial and commercial development, but also provide living and leisure space for urban and rural residents.

It is positively valued the correctness of applied research methods using relevant spatial models to quantify and spatialize and the method of bivariate spatial autocorrelation.

Although the results obtained coincide with those expected according to those that have been achieved in other studies carried out in other territories, it can be affirmed that they are very significant results. In my opinion, it is pertinent to highlight the quantification and identification of temporal differences and territorial contrasts.

The discussion and conclusions are robust and conveniently supported by the results and lead to the affirmation that, given the special location of the urban fringe, making adequate use of land resources is essential for the sustainable development of urban and rural areas.

From a formal point of view, I consider that the exposition of the text is carried out with great clarity and didactic competence, including the cartographic figures that express the results through several series of maps.

Author Response

Thank you very much for your positive and encouraging comments on our manuscript. Your approval will increase our confidence to continue our research. Thanks again for your valuable time, suggestions and comments.